# Antibiotic Prescribing Practices for Treating COVID-19 Patients in Bangladesh

**DOI:** 10.3390/antibiotics11101350

**Published:** 2022-10-04

**Authors:** Utpal Kumar Mondal, Tahmidul Haque, Md Abdullah Al Jubayer Biswas, Syed Moinuddin Satter, Md Saiful Islam, Zahidul Alam, Mohammad Shojon, Shubroto Debnath, Mohaiminul Islam, Haroon Bin Murshid, Md Zakiul Hassan, Nusrat Homaira

**Affiliations:** 1International Centre for Diarrheal Disease Research (icddr,b), 68 Shaheed Tajuddin Ahmed Sarani, Mohakhali, Dhaka 1212, Bangladesh; 2Department of Epidemiology, The Australian National University (ANU), Canberra, ACT 2601, Australia; 3Faculty of Medicine, University of Dhaka, Dhaka 1000, Bangladesh; 4Department of Medicine, Sylhet MAG Osmani Medical University, Sylhet 3100, Bangladesh; 5Discipline of Pediatrics, The University of New South Wales (UNSW), Sydney, NSW 2052, Australia

**Keywords:** COVID-19, antimicrobial resistance, antibiotics, treatment, physicians, Bangladesh

## Abstract

Although national and international guidelines have strongly discouraged use of antibiotics to treat COVID-19 patients with mild or moderate symptoms, antibiotics are frequently being used. This study aimed to determine antibiotics-prescribing practices among Bangladeshi physicians in treating COVID-19 patients. We conducted a cross-sectional survey among physicians involved in treating COVID-19 patients. During September–November 2021, data were collected from 511 respondents through an online Google Form and hardcopies of self-administered questionnaires. We used descriptive statistics and a regression model to identify the prevalence of prescribing antibiotics among physicians and associated factors influencing their decision making. Out of 511 enrolled physicians, 94.13% prescribed antibiotics to COVID-19 patients irrespective of disease severity. All physicians working in COVID-19–dedicated hospitals and 87% for those working in outpatient wards used antibiotics to treat COVID-19 patients. The majority (90%) of physicians reported that antibiotics should be given to COVID-19 patients with underlying respiratory conditions. The most prescribed antibiotics were meropenem, moxifloxacin, and azithromycin. Our study demonstrated high use of antibiotics for treatment of COVID-19 patients irrespective of disease severity and the duty ward of study physicians. Evidence-based interventions to promote judicious use of antibiotics for treating COVID-19 patients in Bangladesh may help in reducing an overuse of antibiotics.

## 1. Introduction

Antimicrobial resistance (AMR) is one of the leading public health issues mostly affecting low- and middle-income countries (LMIC), and the World Health Organization (WHO) declared AMR as one of the ten leading global public health threats [1]. More than 1.25 million people worldwide died in 2019 as a direct effect of infections by antibiotic-resistant bacteria, constituting 26% of the total drug-resistant infections cases (4.95 million) [2]. It has been estimated that by 2050, AMR will be responsible for 10 million preventable deaths annually; furthermore, this will result in an economic loss of 100 trillion US dollars [3,4,5]. This loss will disproportionately impact LMICs, ultimately worsening global poverty and economic inequality [6]. It is estimated that by 2050, the global economic impact of AMR will push back an additional 28.3 million people under the poverty line, the majority of whom (26.2 million) would be in LMICs which would result in more than a 5% loss of the national gross domestic product (GDP) in low resource countries [7].

As of 1 March 2022, the ongoing COVID-19 pandemic has resulted in around 435 million confirmed cases and nearly 6 million deaths globally [8]. Although COVID-19 is primarily a viral infection, different antibiotics are being used frequently to treat COVID-19 patients since the beginning of the pandemic [9,10,11]. Due to the ambiguity of an effective treatment strategy, as an empirical approach, antibiotics are being prescribed to resolute suspected bacterial co-infections [12,13,14,15]. However, reports suggest that the frequency of secondary bacterial infections in COVID-19 patients is very low (<2%) [16]. Prior studies conducted in China and the USA reported that most COVID-19 patients received irrational doses of antibiotics, although very few of them had confirmed bacterial infections [17,18,19]. Moreover, due to the resemblance of COVID-19 symptoms with bacterial pneumonia, physicians often get confused about the source of infection and prescribe broader spectrum antibiotics without any microbiological confirmation of infection to attain better patient compliance [20]. A suspected COVID-19 patient also might have non-specific signs and symptoms like fever or persistent cough which are often mistaken for tuberculosis or malaria, and leads to misuse of antibiotics due to lack of timely and adequate diagnostic measures [21,22]. On top of that, patients’ expectation of well-being around antibiotics, physicians’ fear of losing patient acquisition, the promotion of antibiotics with misleading information, and lucrative monetary incentives provided by pharmaceutical companies often motivate physicians to prescribe costly and unnecessary antibiotics to patients [23]. Such widespread and irrational use of antibiotics clearly indicates the potential emergence of AMR, and accounts for excessive healthcare costs [24,25].

Bangladesh is a densely populated low to middle income country with a total population of 164.6 million [26]. As of the end of February 2022, there have been around two million confirmed cases including about 29,000 deaths, as per Institute of Epidemiology, Disease Control and Research, Bangladesh (IEDCR) estimation (https://dghs-dashboard.com/pages/covid19.php, accessed on 1 April 2022). To ensure proper clinical treatment and management of COVID-19 patients, different national and international clinical guidelines have been developed. According to the WHO guidelines, the use of antibiotics is strongly discouraged to treat COVID-19 patients with mild or moderate symptoms, and should only be used when there is a confirmed secondary bacterial or fungal co-infection [27]. The national guideline on clinical management of COVID-19 treatment also recommends use of antibiotics as per illness severity and only in the presence of co-infections. However, these guidelines are not being routinely followed in many countries across the world [28,29]. There is a paucity of information regarding physicians’ beliefs and practices around the use of antibiotics for treating COVID-19 patents. We conducted a survey to determine antibiotic prescribing practices among Bangladeshi physicians in the treatment of COVID-19 patients. The findings from our study will help develop a consensus for the need of antimicrobial stewardship, and behavioral or psychosocial interventions to improve judicious use of antibiotics in treating COVID-19 patients.

## 2. Material and Methods

### 2.1. Study Design, Period, and Participants

We conducted a cross-sectional study to explore the antibiotic prescribing practices among Bangladeshi physicians who provided clinical care to COVID-19 patients in Bangladesh. Eligible criteria for study participants were: (1) medical doctors (MBBS) registered with the Bangladesh Medical and Dental Council (BMDC) and (2) involved in the treatment of COVID-19 patients. The data were collected from September through November of 2021.

### 2.2. Data Collection

Due to pandemic-related lockdowns and movement restrictions, we adopted two formats for data collection modes [30,31,32]. The self-reported online survey was conducted utilizing the Google Form web survey platform. The survey tool was designed in English, with validation choices such as “mandatory” and “Limited to one answer”. Participants recruitment was carried out through social media invites (Facebook, Messenger, WhatsApp). Before sending the invitation, the study investigators approached six physicians from various locations in Bangladesh (Dhaka, Chittagong, Mymensingh, Sylhet, and Jessore districts) using their professional network to choose who suited the study’s eligibility criteria [33]. These physicians were then further requested to reach out to a broader network of physicians from almost all regions of the country who met the eligibility criteria and might be willing to participate in the study. Out of eight administrative divisions in Bangladesh, physicians from all the divisions (44 districts out of 64) participated in this survey. In addition, study investigators shared Google links on social media forums such as Facebook groups called “Platform”, “Medi voice”, and “Axis”, each having more than 100,000 members, all of whom are Bangladeshi physicians. Once the predetermined sample size was attained, the initially approached six physicians stopped the process of disseminating Google links among their networks. We also used paper hardcopy self-administered questionnaires. The six physicians contacted initially went to their local medical college hospitals, clinics, or doctors’ private practice rooms and approached physicians to participate in the study. Figure 1 shows the data collection survey procedure utilizing online and offline data collection modes.

### 2.3. Survey Instrument

We developed a structured, standardized, and self-administered questionnaire following an extensive literature review [34,35,36,37]. The questionnaires were pre-tested among twenty physicians and were refined based on their inputs (Appendix A). The data from the pilot interviews were excluded from the final data set.

### 2.4. Survey Variables and Measurement

The survey questionnaire consisted of eighteen questions separated into three sections: (a) demographic characteristics (six questions); (b) factors associated with prescribing antibiotics for COVID-19 patients (eleven questions); and (c) types of antibiotics prescribed for treatment of COVID-19 patients according to the severity of the illness (one question). The first section of the survey questionnaire was focused on collecting demographic information about participants, which included participants’ age in years, gender, highest educational level, years of work experience, type of healthcare facility, and duty wards. The second section utilized a 5-point Likert scale (strongly agree to disagree strongly) to collect physicians’ insight into the main decision-making factors in prescribing antibiotics for COVID-19 patients. Physicians’ factors associated with prescribing antibiotics were represented by 11 statements, including three statements on the patient’s underlying medical conditions, three statements on symptoms or signs that influenced physicians to prescribe antibiotics, and five statements on laboratory tests. The third section included names of different types of antibiotics and their usage based on the severity of COVID-19. The severity levels of COVID-19 illness were categorized as mild to moderate, severe, and critical [28,38]. The third section contained different types of antibiotic names in which physicians were asked to choose the antibiotic name for treating COVID-19 patients according to the severity of COVID-19.

### 2.5. Sample Size and Sampling Technique

Assuming 74.6% of patients with COVID-19 are prescribed with antibiotics, with 5% absolute precision, 1.27 design effect, a 10% non-response rate, and using a single proportional approach, we estimated that 511 physicians needed to be surveyed [39]. The total number of registered MBBS physicians in Bangladesh is 109,500 as per the Bangladesh Medical and Dental Council register (https://www.bmdc.org.bd, accessed on 1 April 2022). We could not employ the probability sampling approach, even though we computed the sample size to determine the minimal number of samples needed to produce the true population estimates. The non-probability sample designs were chosen due to the associated risk of COVID-19, physicians’ fear of transmission via contact, incapacity or unwillingness of physicians to continue patient treatment, and increased workplace absenteeism [40]. We used two types of non-probability sampling, including snowball sampling and convenience sampling [30,31]. Figure 1 depicts the detailed survey procedure.

### 2.6. Ethical Consideration

All survey participants gave their consent to participate in the anonymous online survey. Prior to participation in the survey, participants were provided information on the objectives of the study, what was asked for their participation, and risk and benefits associated in this study. After reading this information, they had to check the “Yes” button confirming that they have read and understood the written informed consent form concerning data protection and accepted the regulations. Without checking the “Yes” button, participation was not possible. We strictly ensured the safety, confidentiality, and comfort of the respondents by protecting the privacy of the respondent as per the guideline of American Association for Public Opinion Research (AAPOR) for the protection of human subjects with a study involving survey interviews or questionnaires; therefore, this survey meets the requirement for the exemption of IRB process (https://www.aapor.org/Standards-Ethics/Institutional-Review-Boards/Full-AAPOR-IRB-Statement.aspx, accessed on 1 April 2022). Participation in this survey did not put respondents at more than a minimal risk in their everyday life. Personal identifiable information was not collected, and demographic information were kept separate from the interview data in a secure and password-protected file which only study investigators had access to. For hard copies of the questionnaires, all the participants were informed about the purpose and intent of the study, as well as their requirement for their participation. They were also informed that their participation was voluntary and that they had the right to withdraw their participation at any point in time. Enrollment in this study was done only after obtaining written informed consent. To maintain confidentiality of the participants, personal identifiable data were not collected and other related data that could be linked to a participant were kept confidential to the greatest extent as possible. All the hard copies of the interviews were kept in a locked cabinet and were only used by the researchers involved in the study.

We followed the Checklist for Reporting Results of Internet E-Surveys (CHERRIES) for conducting our online survey [41]. Furthermore, participants were informed about their right to withdraw at any moment and the associated risks and benefits. The Helsinki Declaration of 1975, updated in 2009, was rigorously adhered to by the research team to guarantee that all methods pertinent to this study met the required ethical standards.

### 2.7. Statistical Analysis

We used Excel to enter data from paper questionnaires. The online survey used a Google Form that was linked to a Google spreadsheet, and data were automatically extracted. Following the download of the Google spreadsheet, we combined it with the Excel data and imported the data into STATA for data management and analysis. Antibiotic prescribing frequency was determined separately for each severity level of COVID-19 illness. Dichotomous variables were coded to reflect zero for no antibiotic prescription and one for antibiotic prescribing. Data collected on a five-point Likert scale were condensed to a three-point Likert scale (Disagree, Unsure, Agree) by combining the frequencies of agreed/strongly agreed and similarly for disagreed/strongly disagreed. We employed descriptive statistics to summarize the respondents’ demographic characteristics and other relevant factors. The chi-square or Fisher’s exact test and stepwise Poisson regression with a backward selection algorithm were employed to assess associations between the antibiotic prescribed for treating COVID-19 patients and respondents’ demographic characteristics and influential decision-making variables. Unadjusted prevalence ratios (PR) with a 95% confidence interval (CI) were reported as crude regression model estimates, whereas multivariable regression model outputs were presented as adjusted prevalence ratios (PR) with a 95% CI. All findings were considered significant when the p-value was less than 0.05. Stata 15 software was utilized for all analyses (Stata Corp. 2013. Stata Statistical Software: Release 13. College Station, TX, USA: Stata Corp LP.). 

## 3. Results

### 3.1. Demographic and Professional Profiles of the Participants

Out of the 511 respondents, 346 (67.71%) were males, and 151 (29.55%) were females, and the majority (68.10%) of the physicians were between age 25 to 29 years, with a mean age of 28.81 ± 4.91(Table 1). The majority (57.14%) of the physicians worked at tertiary care hospitals, and 5% of them worked at COVID-19–dedicated facilities. The participants had a mean experience of 3.75 ± 3.95 years as clinical practitioners and the majority of them (82.97%) worked in inpatient/general/COVID-19 wards, with only 4.89% working in intensive care units (ICU) (Table 1).

### 3.2. Physicians’ Antibiotic Prescribing Practices for COVID-19 Patients

Among all the participating physicians, 94.13% prescribed at least one or more antibiotics to COVID-19 patients irrespective of disease severity (mild to moderate, severe, or critical). The use of antibiotics by the physicians ranged between 86.69% for critical COVID-19 patients to 72.21% for mild to moderately ill COVID-19 patients (Table 1).

### 3.3. Factors Influencing Decision Making around Antibiotic Prescribing for Treating COVID-19 Patients

The majority of the study physicians believed that antibiotics should be given to COVID-19 patients based on their underlying medical conditions. Out of total 511 respondents, the majority (90.41%) of them reported that antibiotics should be given to COVID-19 patients with underlying respiratory illness. Similarly, around 76% of the physicians were in favor of providing antibiotics to elderly patients. Approximately half (49%) of the study participants disagreed about using antibiotics for ambulatory COVID-19 patients.

Around 61% of physicians mentioned that antibiotics should be given to patients with a high body temperature (>37.2 °C) [42], diarrhea (51.7%), and secondary bacterial infection (96%). Among all the laboratory tests, the majority of the respondents (90.80%) mentioned that antibiotics should be prescribed to treat COVID-19 patients only after performing a C-reactive Protein (CRP) test (Table 2).

### 3.4. Association between Physicians’ Antibiotic-Prescribing Practices and Associated Factors

Table 3 explores the factors that influence the prescription of antibiotics by physicians. After controlling for confounders, we found that physicians working in the outpatient wards were 1.26 times (95% CI: 1.06–1.51) more likely to prescribe antibiotics than those working in inpatient wards. Moreover, we found certain beliefs, including patients treated outside a healthcare facility should be given antibiotics (adjusted PR 1.23; 95% CI 1.09–1.40) as influential factors for physicians' antibiotic prescribing practices. Similar significant factors were found for the laboratory tests, including a patient's blood hemoglobin level should be checked before giving antibiotics ( adjusted PR 1.20; 95% CI 1.05–1.36); antibiotics should be given after performing a creatinine test ( adjusted PR 1.19; 95% CI 1.03–1.37) (Table 3).

### 3.5. Antibiotic Prescribing Practices

Study participants reported prescribing various types of antibiotics for treating COVID-19 patients irrespective of the bacterial etiology. The most commonly prescribed antibiotics were meropenem, moxifloxacin, and azithromycin. Meropenem and moxifloxacin were the two most common antibiotics used to treat critical and severe COVID-19 patients. Azithromycin was used commonly to treat mild to moderate COVID-19 illness. Figure 2 and Figure 3 illustrate the antibiotic prescribing practices for treatment of COVID-19 patients irrespective of disease severity and the duty ward of the study physicians.

## 4. Discussion

To the best of our knowledge, this is one of the initial surveys assessing the antibiotic prescribing practices among physicians for treating COVID-19 patients in Bangladesh and exploring the associated factors that influenced antibiotic prescribing practices. Our study documented a high frequency (94%) of antibiotic prescribing practices among Bangladeshi physicians for treating COVID-19 patients. Inappropriate and irrational use of antibiotics, particularly during the current COVID-19 pandemic, has evidently become a cross-cultural practice in many countries around the world, which harbors the escalation of antimicrobial resistance to most of the antibiotics that are globally used [43,44].

Antibiotic prescribing practices ranged between 72% and 87% based on severity of patients’ illness (mild to critical), which is comparable to previous studies done in other settings [29,45,46]. There is also evidence that, during the initial phase of the COVID-19 pandemic, every three out of four patients received antibiotic therapy in almost all countries [20]. It is likely that, since any specific guideline for COVID-19 treatment was not yet onboard, until then, physicians started prescribing antibiotics for patients’ symptomatic remedy. The high usage of antibiotics for treating COVID-19 patients identified from our study was not consistent with either the national (which strongly recommends antibiotics use only in critical co-infection conditions) or the WHO’s proposed guidelines of COVID-19 treatment guidelines [27], highlighting an area of improvement for ensuring the optimal use of antibiotics.

Around 90% of the physicians in our study preferred to prescribe antibiotics to COVID-19 patients who had any pre-existing respiratory problem, and 75% of the participants preferred to prescribe antibiotics to elderly patients. These findings are similar to previous studies which also documented a high usage of antibiotic therapy among elderly (median age > 53 years) patients with COVID-19 illness and among those requiring mechanical ventilation [20]. Almost all the physicians (96%) opined in favor of prescribing antibiotics to COVID-19 patients if they had secondary bacterial infections. A considerable number (>50%) of physicians put an emphasis on looking for high fever and gastro-intestinal symptoms (i.e., loose motion) apart from pneumonia as probable symptoms of a bacterial or fungal co-infection while reporting their antibiotic prescription practice, particularly when proper microbiological tests were not timely available [47]. These pertaining beliefs were completely aligned with internationally approved COVID-19 management clinical guidelines that strongly recommended to prescribe antibiotics for COVID-19 patients with suspected bacterial co-infections [27].

Most of the participants (around 91%) preferred to conduct a CRP test prior to prescribing any antibiotic to a COVID-19 patient, as the test count would help them decide the next treatment regime. CRP count is generally elevated in bacterial infections and often helps physicians to differentiate between viral and bacterial infections and thus helps in prescribing required antibiotics [48]. This is identical to the findings reported in two previous studies conducted in 2020, where having a raised value in CRP (≥100 mg/L) among patients influenced the odds of receiving antibiotic therapy [48,49]. As an inexpensive point of care, CRP can help reduce irrational antibiotic prescribing among physicians [50]. 

We also looked at the type of antibiotics being reported to have been commonly prescribed by the study respondents among COVID-19 patients with varying severity. Meropenem (a third-generation antibiotic) and Moxifloxacin (a fluoroquinolone antibiotic) were reportedly frequently prescribed to treat patients with critical and severe COVID-19. As per guidelines provided by the Ministry of Health and Family Welfare (MOHFW) of Bangladesh on the clinical management of COVID-19 (version 8.00) [51], the use of Meropenem has been advocated for among severely ill COVID-19 patients, which probably influenced the study physicians’ decision-making process [29]. However, Azithromycin (broad-spectrum macrolide antibiotic) was reported as the first preference for the majority of the physicians to treat mild COVID-19 patients, although both the national and international guidelines have strongly discouraged the use of any antibiotics in mild COVID-19 illness treatment [27,29].

The high frequency of irrational antibiotic prescribing practices among our study respondents can be subjected to many speculations around the potential underlying causes of such practices. One explanation might be the historic tendencies of Bangladeshi physicians to prescribe antibiotics even for the simplest form of respiratory viral infections and mostly without approaching a diagnostic test [52,53]. On top of that, as the onset of COVID-19 pandemic has been impromptu, there have been fearful propositions about the potential threats of this viral infection getting carried among patients, and the physicians might have found using antibiotics more helpful to be on the safe side and also to make their clienteles satisfied, irrespective of the capability of these drugs to bring remedy to the patients [52,54]. Further, most of the physicians could not avail a detailed guideline of the treatment protocol of COVID-19 patients at the point of care provision, which might have altered their practices compared to the standard guidelines [55]. Although the presence of a functional antimicrobial stewardship program (ASP) is mandatory for every healthcare center of Bangladesh as per the National Action Plan (2017) and Global Antibiotic Resistance Partnership (GARP)-Bangladesh National Working Group Report (2018) [56], their existence is mostly confined to theories. Additionally, there is no established national antibiotic prescribing/ASP guideline let alone the presence of contextualized guidelines in the hospitals except the country’s premiere medical university [57], which puts the physicians in further troubles to align with the concepts of ASP and thus prescribe antibiotics mostly as per their own perceptions. 

Our study has certain limitations. We adopted snowball and convenience sampling which use nonrandom selection procedures mostly depending on subjective judgement of the participants. These sampling techniques may also lead to selection biases which impact generalization, representativeness, and external validation of the findings as the findings might not be consistent for sub-groups of the same population or different populations. The findings of the study might not be constant for subsequent waves of this pandemic as treatment modality for SARS-CoV-2 is rapidly changing. Additionally, data about some clinical variables important for assessing decision making around antibiotic prescribing were not collected (i.e., necessity of procalcitonin test, D Dimer, neutrophil counts, differentiating signs among viral and bacterial infections, prescribing practice for beta-lactamase inhibitors). Another issue is the utilization of the Google Form and recruitment of participants through social media invites (Facebook, Messenger, WhatsApp) that might have been a barrier for older-aged physicians for participating in the study. However, this study still provides comprehensive data on antibiotic prescribing practices among physicians for treating COVID-19 patients. We also tried to capture the diversity in practices through enrolling participants from different settings including ambulatory care and inpatient settings.

## 5. Conclusions

We have demonstrated high antibiotic prescribing practices among physicians in Bangladesh for treating COVID-19 patients, specifically COVID-19 patients with mild illness where antibiotics are not recommended by local guidelines. Such blanket use of antibiotics during the pandemic may contribute to the emergence of AMR. The study findings also highlight the need for context-specific feasible interventions for promoting, strengthening, and sustaining the anti-microbial stewardship program (ASP) to ensure judicious use of antibiotics. 

## Figures and Tables

**Figure 1 antibiotics-11-01350-f001:**
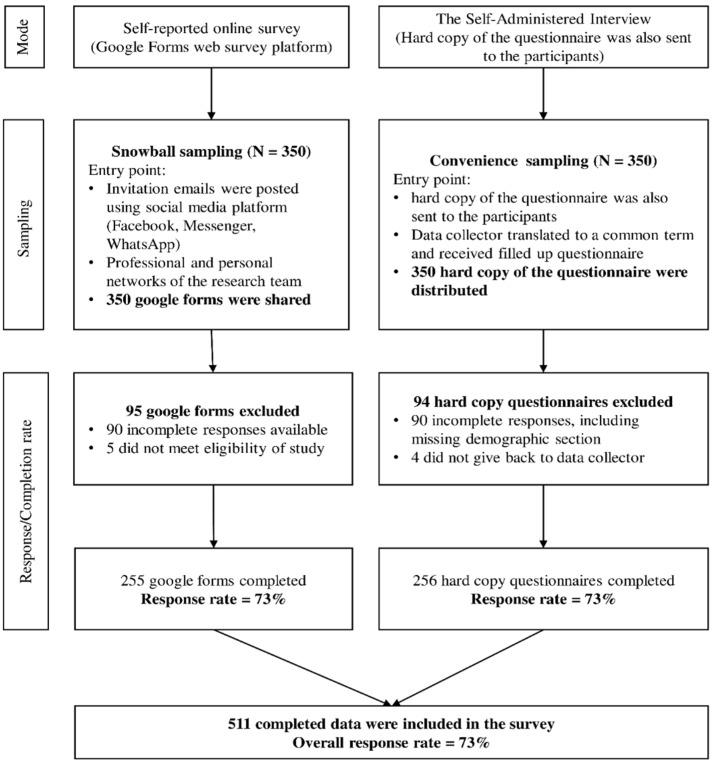
Flowchart depicting the data collection survey procedure using two sampling techniques.

**Figure 2 antibiotics-11-01350-f002:**
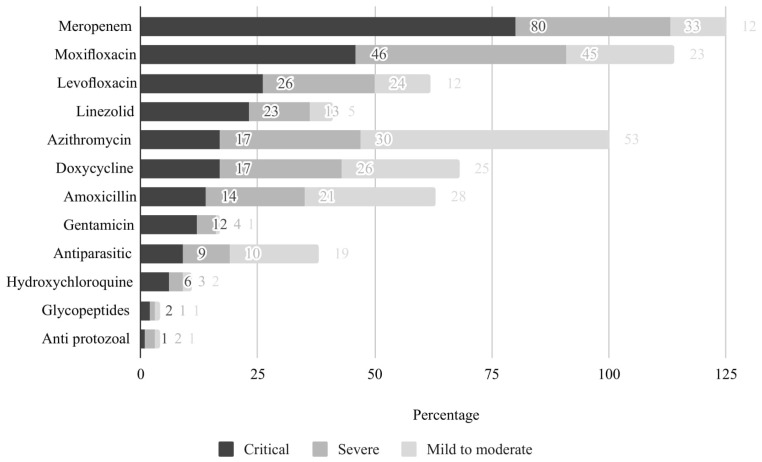
Antibiotic prescribing practices among the physicians for all three COVID-19 illness categories.

**Figure 3 antibiotics-11-01350-f003:**
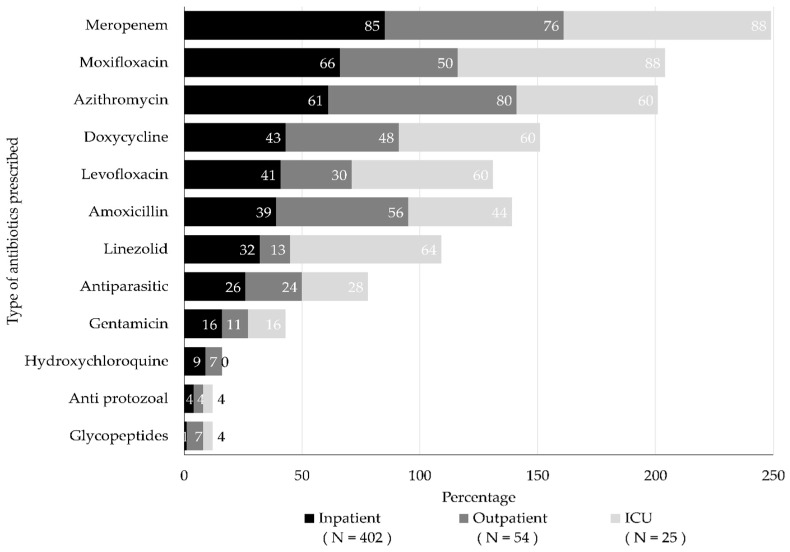
Type of antibiotics prescribed to CVOID-19 patients, regardless of illness severity, based on duty ward of the study physicians.

**Table 1 antibiotics-11-01350-t001:** Demographic characteristics of the study physicians, 2021 Bangladesh.

Background Characteristics	All Study Samples	Severity Level of COVID-19 Illness				
		Mild to Moderate	Severe	Critical	Mild to Critical	
	col% (*n*)	row% (*n*)	row% (*n*)	row% (*n*)	row% (*n*)	*p*-Value
**Overall**	100 (511)	72.21 (369)	78.67 (402)	86.69 (443)	94.13 (481)	
**Age in year**						
Mean ± SD	28.81 ± 4.91	28.92 ± 4.77	29.09 ± 5.01	29.00 ± 4.69	29.01 ± 4.81	**<0.001**
≤24	3.52 (18)	22.22 (4)	50.00 (9)	72.22 (13)	72.22 (13)	
25–29	68.10 (348)	75.29 (262)	79.60 (277)	84.77 (295)	93.68 (326)	
30–34	18.59 (95)	75.79 (72)	77.89 (74)	95.79 (91)	97.89 (93)	
≥35	9.78 (50)	62.0 (31)	84.00 (42)	88.00 (44)	98.00 (49)	
**Gender**						
Male	67.71 (346)	66.23 (100)	70.86 (107)	86.09 (130)	90.07 (333)	**0.001**
Female	29.55 (151)	74.86 (259)	82.37 (285)	86.99 (301)	96.24 (136)	
Prefer not to disclose	2.74 (14)	71.43 (10)	71.43 (10)	85.71 (12)	85.71 (12)	
**Education**						
Graduate	87.87 (449)	73.50 (330)	79.06 (355)	85.52 (384)	93.76 (421)	0.345
Post-graduate	12.13 (62)	62.90 (39)	75.81 (75.81)	95.16 (59)	96.77 (60)	
**Experience in Years as a Clinical Practitioner**						
Mean ± SD	3.75 ± 3.95	3.77 ± 3.90	3.90 ± 4.11	3.89 ± 4.11	3.84 ± 4.0	0.056
<3	56.56 (289)	71.63 (207)	77.16 (223)	84.43 (244)	92.39 (267)	
≥3	43.44 (222)	72.97 (162)	80.63 (179)	89.64 (199)	96.40 (214)	
**Type of Healthcare Setting ***						
Tertiary	57.14 (292)	71.58 (209)	82.88 (242)	87.67 (256)	95.55 (279)	0.115
Primary	11.35 (58)	74.14 (43)	70.69 (41)	91.38 (53)	94.83 (55)	0.810
Secondary	9.98 (51)	68.63 (35)	74.51 (38)	84.31 (43)	88.24 (45)	0.059
COVID-19–dedicated	4.95 (27)	81.48 (22)	92.59 (25)	92.59 (25)	100.00 (27)	0.182
Private hospitals	21.47 (117)	71.79 (84)	83.76 (98)	83.76 (98)	92.31 (108)	0.340
**Duty Ward**						
Inpatient + general ward + COVID-19 ward	82.97 (424)	71.23 (302)	79.72 (338)	86.56 (367)	94.81 (402)	**0.024**
Outpatient	12.13 (62)	72.58 (45)	69.35 (43)	82.26 (51)	87.10 (54)	
ICU	4.89 (25)	88.00 (22)	84.00 (21)	100.00 (25)	100.00 (25)	

* Multiple responses (participants worked in more than one setting).

**Table 2 antibiotics-11-01350-t002:** Factors influencing decision making around prescribing antibiotics for treating COVID-19 patients, 2021 Bangladesh.

Influential Decision-Making Factors		All Study Samples	Severity Level of COVID-19 Illness				
			Mild to Moderate	Severe	Critical	Mild to Critical	
		col% (*n*)	row% (*n*)	row% (*n*)	row% (*n*)	row% (*n*)	*p*-Value
**Patient’s underlying medical conditions**							
Patients with respiratory illnesses should be given antibiotics	Agree	90.41 (462)	75.76 (350)	80.74 (373)	88.10 (407)	96.10 (444)	**<0.001**
	Unsure	4.89 (25)	32.00 (8)	56.00 (14)	72.00 (18)	72.00 (18)	
	Disagree	4.70 (24)	45.83 (11)	62.50 (15)	75.00 (18)	79.17 (19)	
In the case of an elderly patient, antibiotics should be given	Agree	75.73 (387)	76.74 (297)	80.86 (311)	87.34 (338)	95.87 (371)	**0.002**
	Unsure	11.35 (58)	65.52 (38)	77.59 (45)	89.66 (52)	93.10 (54)	
	Disagree	12.92 (66)	51.52 (34)	69.70 (46)	80.30 (53)	84.85 (56)	
Patients who are treated outside of a healthcare facility should be given antibiotics	Agree	28.38 (145)	83.45 (121)	89.66 (130)	93.10 (135)	99.31 (144)	**0.007**
	Unsure	22.70 (116)	70.69 (82)	73.28 (85)	86.21 (100)	91.38 (106)	
	Disagree	48.92 (250)	66.40 (166)	74.80 (187)	83.20 (208)	92.40 (231)	
**Signs/Symptoms**							
Patients with a high temperature should be given antibiotics	Agree	61.45 (314)	76.43 (240)	83.44 (262)	91.40 (287)	98.73 (310)	**<0.001**
	Unsure	16.24 (83)	74.70 (62)	71.08 (59)	75.90 (63)	86.75 (72)	
	Disagree	22.31 (114)	58.77 (67)	71.05 (81)	81.58 (93)	86.84 (99)	
Patients with loose motion should be given antibiotics	Agree	51.66 (264)	77.27 (204)	80.68 (213)	89.02 (235)	98.11 (259)	**<0.001**
	Unsure	16.63 (85)	74.12 (63)	74.47 (65)	84.71 (72)	84.41 (76)	
	Disagree	31.70 (162)	62.96 (102)	76.54 (124)	83.65 (136)	90.12 (146)	
If the patients have a secondary bacterial infection, they should be given antibiotics.	Agree	96.09 (491)	72.91 (358)	79.23 (389)	87.17 (428)	94.70 (465)	**0.002**
	Unsure	2.94 (15)	60.00 (9)	73.33 (11)	73.33 (11)	73.33 (11)	
	Disagree	0.98 (5)	40.00 (2)	40.00 (2)	80.00 (4)	100.0 (5)	
**Laboratory Tests**							
Before giving antibiotics, a patient’s blood hemoglobin level should be checked	Agree	44.42 (227)	80.62 (183)	80.18 (182)	87.22 (198)	96.48 (219)	**0.045**
	Unsure	18.40 (94)	73.40 (69)	79.79 (75)	84.04 (79)	89.36 (84)	
	Disagree	37.18 (190)	61.53 (117)	76.32 (145)	87.37 (166)	93.68 (178)	
Antibiotics should be given after performing a creatinine test	Agree	66.54 (340)	77.06 (262)	79.71 (271)	88.53 (301)	95.88 (326)	**0.018**
	Unsure	11.94 (61)	72.13 (44)	77.05 (47)	83.61 (51)	86.89 (53)	
	Disagree	21.53 (110)	57.27 (63)	76.36 (84)	82.73 (91)	92.73 (102)	
Antibiotics should be given after determining the bilirubin level	Agree	36.79 (188)	76.06 (143)	78.72 (148)	88.83 (167)	97.34 (183)	**0.018**
	Unsure	27.01 (138)	71.01 (98)	77.54 (107)	84.06 (116)	89.86 (124)	
	Disagree	36.20 (185)	69.19 (128)	79.46 (147)	86.49 (160)	94.05 (174)	
Antibiotics should be given after performing an ALT test	Agree	50.29 (257)	75.10 (193)	81.32 (209)	91.05 (234)	97.67 (251)	**0.001**
	Unsure	23.48 (120)	70.00 (84)	74.17 (89)	81.67 (98)	88.33 (106)	
	Disagree	26.22 (134)	68.66 (92)	77.61 (104)	82.84 (111)	92.54 (124)	
Antibiotics should be given after a CRP test has been performed	Agree	90.80 (464)	74.35 (345)	79.74 (370)	88.15 (409)	95.69 (444)	**<0.001**
	Unsure	5.68 (29)	58.62 (17)	68.97 (20)	72.41 (21)	75.86 (22)	
	Disagree	3.52 (18)	38.89 (7)	66.67 (12)	72.22 (13)	83.33 (15)	

**Table 3 antibiotics-11-01350-t003:** Bivariable and multivariable regression analysis to explore factors associated with the overall prevalence of prescribing antibiotics for treating COVID-19 patients.

Background Characteristics		Unadjusted PR (95% CI)	*p*-Value	Adjusted PR (95% CI)	*p*-Value
**Age in Year**					
≤24		1.74 (1.24–2.45)	**0.001**	-	
25–29		1.62 (1.14–2.30)	**0.007**	-	
30–34		1.43 (0.99–2.67)	0.058	-	
≥35		Reference			
**Type of Working Ward**					
ICU		0.88 (0.70–1.13)	0.324	0.89 (0.71–1.11)	0.309
Outpatient		1.31 (1.08–1.60)	**0.007**	1.26 (1.06–1.51)	**0.01**
Inpatient		Reference		Reference	
**Determinant for Antibiotic Use**					
**1. Patients’ Underlying Medical Conditions**					
Patients with respiratory illnesses should be given antibiotics	Unsure	0.58 (0.39–0.86)	**0.006**	0.70 (0.50–0.99)	**0.046**
	Disagree	0.56 (0.40–0.77)	**0.001**	0.75 (0.53–1.05)	0.095
	Agree	Reference		Reference	
In the case of an elderly patient, antibiotics should be given	Unsure	0.86 (0.72–1.03)	0.088	-	
	Disagree	0.61 (0.50–0.75)	**<0.001**	-	
	Agree	Reference			
Patients who are treated outside of a healthcare facility should be given antibiotics	Agree	1.44 (1.27–1.64)	**<0.001**	1.23 (1.09–1.40)	**0.001**
	Unsure	1.04 (0.89–1.22)	0.571	0.97 (0.84–1.12)	0.704
	Disagree	Reference		Reference	
**2. Signs/Symptoms**					
Patient with a high fever should be given antibiotics	Agree	1.42 (1.21–1.67)	**<0.001**	1.13 (0.96–1.32)	0.122
	Unsure	1.21 (0.97–1.49)	0.088	1.04 (0.85–1.26)	0.714
	Disagree	Reference		Reference	
Patient with loose motion should be given antibiotics	Agree	1.31 (1.14–1.49)	**<0.001**	-	
	Unsure	1.18 (0.98–1.42)	0.087	-	
	Disagree	Reference			
If the patients have a secondary bacterial infection, they should be given antibiotics	Unsure	0.92 (0.60–1.41)	0.705	1.09 (0.73–1.63)	0.662
	Disagree	0.32 (0.20–0.53)	**<0.001**	0.38 (0.27–0.56)	**<0.001**
	Agree	Reference		Reference	
**3. Laboratory Tests**					
Before giving antibiotics, a patient’s blood hemoglobin level should be checked.	Agree	1.40 (1.24–1.59)	**<0.001**	1.20 (1.05–1.36)	**0.005**
	Unsure	1.46 (1.15–1.61)	**<0.001**	1.24 (1.05–1.46)	**0.013**
	Disagree	Reference		Reference	
Antibiotics should be given after performing a creatinine test	Agree	1.47 (1.27–1.69)	**<0.001**	1.19 (1.03–1.37)	**0.019**
	Unsure	1.34 (1.08–1.66)	**0.007**	1.13 (0.91–1.41)	0.262
	Disagree	Reference		Reference	
Antibiotics should be given after determining bilirubin level	Agree	1.24 (1.09–1.42)	**0.001**	-	
	Unsure	1.14 (0.98–1.33)	0.091	-	
	Disagree	Reference			
Antibiotics should be given after performing an ALT test first	Agree	1.24 (1.07–1.43)	**0.003**	-	
	Unsure	1.10 (0.92–1.32)	0.278	-	
	Disagree	Reference			
Antibiotics should be given after a CRP test has been performed	Agree	1.54 (1.04–2.31)	**0.032**	-	
	Unsure	1.48 (0.84–2.59)	0.177	-	
	Disagree	Reference			

## Data Availability

The data presented in this study are available on request from the corresponding author. The data are not publicly available due to maintaining privacy.

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
