# Peer review of "Antibiotic Prescribing Practices for Treating COVID-19 Patients in Bangladesh"

_antibiotics, 2022, doi:10.3390/antibiotics11101350_

Round 1

Reviewer 1 Report

This is a study conducted in Bangladesh that evaluates antibiotic prescriptions for patients with covid-19 through an electronic questionnaire. The study is well designed and interesting to the readers . However, some points need to be clarified.

Abstract

Line 28-29

“The most prescribed antibiotics were meropenem, moxifloxacin and azithromycin. Our 28 study demonstrated high use of antibiotics for treatment of COVID-19 patients irrespective of dis- 29 ease severity.”

It will be interesting to added information regarding the use of antibiotic in outpatients and in patients.

Introduction

The introduction can be shortened. I suggested to excluded or shortened the first paragraph.

 Material and Methods

92 2.1. Study design, period, and participants

Lines 120:  Survey Instrument

Could the authors add the survey instrument as supplementary data.

Lines

Although the authors calculated the power and size of the samples of participants. It will be interesting to have more detail about the representative of the participants.

How many physicians there are in Bangladesh? The study was conducted in only one region of the country?

The authors described the following phrase: “Before sending the invitation, the study investigators approached physicians from various locations in Bangladesh using their professional network who suited the study's eligibility criteria (35). Only six of the approached physicians from Dhaka, My- mensingh, Chittagong, Jessore and Sylhet districts consented to participate in the study voluntarily.

However, it is not clear the representative of the sample of the study.  

Please correct the mistyping as table 2 Aantibiotic.

Results

The age of participants ranged from to, thus only young physician answered the questionnaire. Do you think that the use of googleforms and recruitment through social media invites (Facebook, Messenger, 105 WhatsApp) could be a barrier to older physicians answered the questionary?  

If so, could the authors add a phrase about this potential limitation on discussion section.

Figure 2- It would be interesting to add information in the graph on antibiotic prescription for outpatients and inpatients as well.

Discussion

An important issue is to clarify the antibiotics prescription rules in the country. For example  in hospitals, is antibiotic stewardship mandatory?

Conclusion

In the conclusion the authors described: “We have demonstrated high antibiotic prescribing practices among physicians in 7 Bangladesh for treating COVID-19 patients, specifically COVID-19 patients with mild ill- ness where antibiotic is not recommended by local guideline.”

 It is important to emphasize this finding in the discussion and hypothesize the reason for such finding.

This is a study conducted in Bangladesh that evaluates antibiotic prescriptions for patients with covid-19 through an electronic questionnaire. The study is well designed and careful. However, some points need to be clarified.

Reviewer 2 Report

Thank you for offering for peer review this study describing antibiotic prescribing practices for treatment COVID-19 in Bangladesh. They used a survey approach to capture responses from doctors. There is increasing literature around this topic however the context of practice in Bangladesh, a LMIC, is of interest.

There are numerous syntax errors identified throughout, I have listed a few here however this is not a comprehensive list and I suggest authors thoroughly re-read the submission to ensure all are identified and corrected.

The following are some suggestions to consider

·      Authors to look at consistency of aims as described in title, abstract and body noting the interchangeable use of terms “practice” and “patterns”. I would suggest following the title of the paper “antibiotic prescribing practices” which will also encompass “patterns”.

·      Introduction paragraph 2 lines 57-73. This paragraphed could be organised more clearly to define the issues that drive prescribing that is different to guidelines (1) concern regarding bacterial co-infection (2) uncertainty as to best treatment strategy/ guidelines

·      Line 90 “judicial” should be “judicious”

·      Data collection and no Ethics – I interpret that the authors are making a case that Google forms can be used and Ethics waived on the basis that the no highly sensitive information is being collected and that the recruitment process was informed, clear consent with no coercion.  They use the American Association for Public Opinion Research Standards as a reference document.  This seems reasonable, however I suggest additional wording in (1) ethical considerations section that provides further detail on identification, recruitment, how prior consent was achieved and anonymity preserved in the self-administered hard copy interviews that were administered (2) the methods section that provides reassurance that no potentially identifiable demographic information was collected

·      Section 3.2 is titled "antibiotic prescribing prevalence" among the physicians.  I would suggest that a clearer title is “physician antibiotic prescribing patterns for COVID-19 patients ”.

·      Table 2 has syntax issues in rows 1 and 3, the questions should also be similar to text e.g. Row 1, column 1 question is different to line 221 where there is a difference in the terms used  “illnesses” versus “conditions”.

·      3.4. In title "antibiotic prescribing prevalence" is used, please see my suggestions for section 3.2 above regarding wording.

·      Table 3.  Abbreviation “PR” should be explained.

·      Line 243 “Regrading” should be “regarding”.

·      Section 3.5 Antibiotic prescribing pattern.  I note the absence of cephalosporins and beta-lactam/beta-lactamase inhibitors (e.g. piperacillin-tazobactam) from the list and linezolid is 4th.  Can I confirm that this is correct? If so, it would be worth elaborating on reasons why in the discussion as these patterns of prescribing would be very different to what is seen in other countries when prescribing for respiratory infections.

·      Line 287 “emphasize” should be “emphasis”. Line 289-290 syntax issues.

·      Paragraph starting 294.  I suggest that wording of findings be modified to less reflect actual practice but rather reflect “reported practice”; I would like to ask for clarification from authors whether the question asked survey respondents indicate that they would be guided to prescribe antibiotics if CRP result available and >100 or was the question phrased such that what is interpreted is that they would like a CRP to have been done because it would be helpful to guide them what to do?

·      As I mentioned in the results section, antibiotic choice including spectrum amongst prescribers, is a finding of interest. I would suggest expanding on this in the discussion.  I am uncertain as to what local guidelines would recommend for respiratory infections, mild, moderate or severe apart from the brief commentary around meropenem for severely ill COVID-19 patients.  Some further commentary on whether this reflects broader-spectrum use than would be otherwise be seen for bacterial pneumonia without COVID-19 would be welcomed.   I gather that the presence/ absence of ASP in the hospitals that respondents worked at was not captured? A brief commentary on ASP in Bangladesh would be of interest.
